# Well-Being, Protein-Bound Toxins, and Dietary Fibre in Patients with Kidney Disease: Have We Been Missing the Obvious?

**DOI:** 10.3390/toxins17110548

**Published:** 2025-11-04

**Authors:** Aruni Malaweera, Louis L. Huang, Lawrence P. McMahon

**Affiliations:** 1Department of Renal Medicine, Eastern Health, 5 Arnold Street, Box Hill, VIC 3128, Australia; louis.huang@monash.edu (L.L.H.); lawrence.mcmahon@monash.edu (L.P.M.); 2Department of Renal Medicine, Eastern Health Clinical School, Monash University, 5 Arnold Street, Box Hill, VIC 3128, Australia

**Keywords:** protein-bound uraemic toxins, adequacy, kidney disease, physical, dialysis

## Abstract

Aim: To explore the associations between protein-bound uraemic toxins (PBTs), fibre intake and patient-focused outcomes in patients on kidney replacement therapy. Background: Despite removal of small water-soluble uraemic toxins, dialysis patients continue to experience high morbidity and mortality. Recent evidence suggests strong associations between PBTs and poorer patient outcomes and symptom burden. Reducing the generation of PBTs by increasing dietary fibre may be an alternate approach to better patient outcomes. Method: This was a cross-sectional study of haemodialysis (HD), peritoneal dialysis (PD) and kidney transplant patients to determine the associations between uraemic toxins [p-cresyl sulfate (PCS) and indoxyl sulfate (IS)], fibre intake and patient-focused outcomes, incorporating the Integrated Palliative Outcome Scale-Renal (IPOS-renal) and EQ-5D-5L to determine symptom burden and quality of life, while physical capacity was determined using the timed up and go(TUG) test and handgrip strength (HGS). Results: Ninety participants completed the study (n = 30 in each group). There was a correlation between PBTs and the IPOS-renal score, where higher toxin levels were associated with a higher symptom burden. This was the strongest for PCS, where the significance remained after accounting for age and co-morbidities (*p* < 0.05). Higher PBT levels were also associated with lower HGS (*p* < 0.05). There was a negative correlation between fibre intake and PBTs, serum PCS (r = −0.36, *p* < 0.05) and serum IS (r = −0.27, *p* < 0.05). Lower fibre intake was also associated with a higher symptom burden measured by the IPOS-renal (*p* < 0.05). Transplant patients consistently performed better, with a reduced symptom burden and improved physical ability compared to dialysis patients. Conclusion: PBTs were associated with symptom burden, and lower physical ability was associated with both PBTs and patient-focused outcomes, and this needs to be further investigated in larger studies.

## 1. Background

Kidney failure leads to the accumulation of uraemic toxins that comprise small water-soluble molecules (<500 Da), middle molecules (>500Da) and protein-bound uraemic toxins (PBTs) [1]. Dialysis aims to restore homeostasis by removing uraemic toxins, according to international clearance targets derived from small water-soluble uraemic toxins such as urea and creatinine [2,3]. Despite this, dialysis patients have significant cardiovascular mortality, where achieving these small-solute clearance targets has shown no correlation with improved patient outcomes [4,5,6]. Therefore, guidelines recommend moving away from small-solute clearance and focusing on other aspects of patients’ care, including patient-focused outcome measures and nutrition [7]. However, it is unclear whether we are measuring the correct uraemic toxins, and whether the increased mortality is due to the accumulation of other uraemic toxins like PBTs, a concept known as ‘additive toxicity’ [8]. PBTs consist of indoxyl-sulfate (IS) and p-cresyl sulfate (PCS) and are produced through metabolism of amino acids by gut bacteria [9,10]. There are multiple cellular, animal and human studies demonstrating the deleterious effects of PBTs, namely nephrotoxicity and cardiovascular mortality [8]. However, only a handful of studies have shown an association of PBTs with patient-focused outcomes. For example, a cross-sectional study by Bammens et al. found that the serum levels of PCS were associated with increased uraemic symptoms, measured by a tool derived from a dyspepsia questionnaire [11]. This is the first study showing an association between PBTs, uraemic symptom burden and other aspects of patient well-being, such as physical capacity or quality of life (QOL).

Nonetheless, given these potential deleterious effects, efforts have been made to determine how best to remove PBTs from the serum. They are poorly removed during dialysis due to strong, non-covalent binding to albumin. Therefore, PBT clearances in HD and PD are 10-fold lower than clearances of urea and/or creatinine [12]. Absorption agents have also been trialled to reduce PBT concentrations, although with limited efficacy and significant side effect profile, which reduce their ability to be more widely adopted in clinical practice [13,14]. Fibre intake has been a recent innovation to reduce PBT production in kidney failure. It has been shown to reduce serum PBT levels in both chronic kidney disease (CKD) and HD patients. Regardless of the type of fibre, its supplementation has been found to reduce serum PBT levels by up to 30% in CKD and HD studies; however, few PD patients have been studied [15,16,17]. Furthermore, it is unclear whether fibre-induced PBT reduction correlates with improved patient outcomes and symptoms.

The aim of this study was to investigate the associations between PBTs, patient-focused outcome measures and fibre intake.

## 2. Results

One hundred patients were approached for the study, and 90 patients completed the study, with 30 in each group. The general characteristics of the cohort are detailed in Table 1. The median age of the group was 63 (IQR 54–74), and most patients were male (n = 59, 66%). The most common causes of end-stage kidney disease were diabetes, hypertension and glomerulonephritis. The median Charlson co-morbidity score was 5 (IQR 3–7).

### 2.1. The Association Between PBTs and Patient-Reported Outcome Measures (PROMs)

There was a statistically significant positive correlation between PCS levels and IPOS-renal scores in a univariable analysis (r = 0.40, *p* < 0.0001) (Figure 1). This remained statistically significant when a multilevel mixed-effects linear regression was performed to account for confounders that may have affected symptom burden (e.g., age and comorbidities) (Table 2). Although there was a statistically significant positive correlation between IS levels and IPOS-renal scores, this lost significance after a regression analysis (Table 3). There were no statistically significant associations between PBTs and the EQ-5D-5L index value, nor with the visual analogue scale (Table 2 and Table 3).

### 2.2. The Association Between PBTs and Physical Ability

There was a negative correlation between HGS and PCS by univariable analysis (r = −0.21, *p* < 0.05) (Figure 1). However, this did not remain significant after adjusting for relevant confounders (Table 2). There was a negative correlation between HGS and IS in both the univariable (r = −0.34, *p* < 0.05) and multivariable analyses (Table 3).

TUG time positively correlated with both PCS and IS levels in the univariable analysis (r = 0.34, *p* < 0.05 for PCS and r = 0.30, *p* < 0.01 or IS). However, these failed to remain significant after performing multilevel mixed-effects linear regression for both PBTs (Table 2 and Table 3).

### 2.3. The Association Between Fibre Intake, PBTs and Patient-Focused Outcome Measures

There was a negative correlation between fibre intake and serum PCS levels (r = −0.36, *p* < 0.01) (Figure 1). Similar results were also found with IS (r = −0.27, *p* < 0.05). Similarly, there was a significant negative correlation between fibre intake and IPOS-renal scores in the univariable analysis (r = 0.39, *p* < 0.001) (Figure 2). This remained significant when a multilevel mixed-effects linear regression was performed (Table 4). In addition, when we reviewed the fibre intake for the entire group, the median split of the fibre intake was 14 g. We also assessed outcomes based on this dichotomy and found that those in the lower fibre group (<14 g/day) had a higher IPOS-renal score or higher symptom burden than the lower fibre group (*p* < 0.05).

Although lower fibre intake was also associated with a longer TUG time in a univariable analysis, it did not hold significance in a multilevel mixed-effects linear regression (r = −0.28, *p* < 0.05) (Table 4). There was no association between fibre intake and HGS and EQ-5D-5L (Table 4).

### 2.4. The Association Between Albumin and Patient-Focused Outcomes

There was a negative correlation between serum albumin and uraemic symptoms (r = −0.32, *p* < 0.01) (Table 5). This remained significant when multilevel mixed-effects linear regression was performed accounting for factors that may affect symptoms (patient demographics) or diet (Table 5). Conversely, there was a positive correlation between serum albumin and HGS (r = 0.32, *p* < 0.01) which remained significant after performing a multilevel mixed-effects regression. Lower serum albumin levels were also associated with poorer quality of life (measured by EQ-5D-5L index value) and longer TUG time in a univariable analysis, but this failed to remain significant after performing a multilevel mixed-effects regression.

### 2.5. Comparison Between Transplant, HD and PD Patients

The general characteristics in the three patient groups are shown in Table 1. The HD group was the oldest compared to the other two groups [median age 72 (IQR 59–80), *p* < 0.05). The HD group also had a higher Charlson co-morbidity score [median score 7 (IQR 5–8), *p* < 0.01] compared to the transplant and PD groups [median score 4 (IQR 3–6)]. The HD group was more likely to use a mobility aid outside compared with transplant patients (*p* < 0.01).

### 2.6. Comparison of Patient-Focused Outcome Measures Between Transplant, HD and PD Patients

The transplant group performed better in most aspects of patient-focused outcome measures compared to dialysis patients (Table 1). The median EQ-5D-5L visual analogue scale was highest in the transplant group [85% (80–90)] compared with the other two groups, where a higher percentage indicates a higher measure of their perception of their overall health. There were no statistically significant differences between the three groups for EQ-5D-5L index value nor there was a difference in QOL measures between PD and HD groups. The median IPOS-renal total score was lower in the transplant [7 (IQR 3–16)] group compared to the other two groups, indicating a lower symptom burden. There was no significant difference in the IPOS-renal scores between the HD and PD groups.

The transplant group had the highest median HGS of 33 kg (IQR 28–45) compared with the PD [24 kg (IQR 18–31), *p* < 0.001) and HD groups [26 kg (IQR 18–34), *p* < 0.001). There was no significant difference in HGS between the HD and PD groups. Similarly, the transplant group had the fastest TUG test, with a median time of 9.4 s (IQR 7.5–10.6) compared to the other groups. There was no difference between the HD and PD groups in TUG times.

### 2.7. Comparison of PBTs and Fibre Intake Between Transplant, HD and PD Patients

PBT levels were lowest in the transplant group compared to the PD and HD groups (Table 1). The median IS level was significantly lower in the transplant group [0 μM (IQR 0–0.6)] compared to the PD [9 μM (IQR 5.1–15.4), *p* < 0.0001) and HD groups [8.4 (IQR 5.0–16.4), *p* < 0.0001].

Similarly, the median PCS level was lower in the transplant group [4.2 μM (IQR 1.2–15.1)] compared to the other two groups.

The median fibre intake in the transplant group was 19.3 g/day (IQR 13.5–26.3) which was significantly higher than the PD group [11.8 g/day (IQR 8.3–15.7), *p* < 0.05] and the HD group [13.0 g/day (IQR 9.9–19.7), *p* < 0.05].

### 2.8. The Effect of Residual Kidney Function

We assessed the effect of residual kidney function on patient-focused outcomes in the HD and PD groups. Low urine output was defined as <250 mL/day or anuria. There were 23 such patients, leaving 37 patients with a urine output of >250 mL/day. There were no significant differences in PBT levels between the two groups. The group with the higher urine output had a higher median EQ-5D-5L VAS score [75% (IQR 65–85)] compared to the group with the lower urine output [60% (IQR 50–80), *p* < 0.05]. We found no differences in total IPOS-renal scores, EQ-5D-5L index values, TUG or HGS between the two groups.

## 3. Discussion

PBTs induce multiple negative systemic biological effects but little is known about their contribution to uraemic symptomatology and patient-focused outcomes. Fibre has been shown to reduce PBT levels in patients with kidney disease, but it is unclear whether this leads to improved patient outcomes. Our study evaluates this complex interplay between PBTs, patient-focused outcomes and fibre intake.

In this study, we found a positive correlation between PBTs and symptom burden, where a higher serum PBT level was associated with a higher symptom burden, especially for PCS. A study by Bammens and co-workers also found similar results, where higher serum PCS concentrations were associated with an increased uraemic symptom burden [11]. However, the questionnaire used was not validated in kidney disease and further studies since have not corroborated these findings.

Our study is the first study to determine the association between PBTs and symptom burden using validated PROM instruments. The assessment of symptom burden in kidney disease is key to assessing patients’ well-being, together with disease impact and functionality [18]. It allows the early identification of concerns and engagement, facilitating shared decision making in establishing care goals. While numerous studies have shown a poor correlation with symptom burden and small water-soluble uraemic toxins, PBTs may align better and contribute to the pathophysiology of uraemic symptoms [19,20,21,22]. Thus, PBTs and their association with uraemic symptoms may allow us to monitor patients’ progress on dialysis. The exact mechanism by which PBTs lead to uraemic symptoms is unclear but could be multifactorial. Alteration of the gut microbiome by overgrowth of harmful bacteria in kidney disease can lead to both the increased production and intestinal modification of PBTs [23]. In addition, the systemic inflammatory state in CKD leads to an increased absorption of PBTs into serum through the concept known as “leaky gut”. CKD also leads to reduced protein-binding, resulting in increased free PBUT levels in the serum [24]. Therefore, in conjunction with the reduced clearance of PBTs through dialysis, there is an accumulation of PBTs in the serum. This leads to oxidative stress, alteration of the blood–brain barrier and neuromodulation that can result in multiple uraemic symptoms.

In addition to relating to symptom burden, we found that PBTs are also negatively associated with physical ability, where higher serum concentrations were associated with lower physical ability. This was strongest for IS and HGS, where the statistical significance remained after accounting for confounders that may affect muscle strength. The association of HGS and PBTs has been studied in the past with mixed results. Similarly to our study, Hou and Lin both found that higher serum IS levels were associated with a lower HGS in both cross-sectional and longitudinal studies [25,26]. However, the data are mixed. Other studies in HD patients have failed to show an association between PBTs, muscle strength and sarcopenia [27,28]. Additional trials investigating change in physical strength with PBTs may help further address this relationship.

In this study, we also found that higher fibre intake was associated with lower serum PBTs. This is similar to studies in CKD and HD patients, where a higher fibre intake led to a reduction in PBTs, possibly by improving gastric transit and reducing toxin absorption, improving the integrity of tight junctions in the colonic epithelium and/or facilitating a more favourable microbiome [8]. However, it is unclear whether the reduction in PBTs with fibre intake improves patient outcomes. In our study, we found that lower fibre intake was associated with longer TUG and a higher IPOS-renal score, where IPOS-renal score proved to be an independent variable that may affect symptoms and muscle strength. To date, studies show fibre reduces gastro-intestinal symptoms and improves sleep quality in patients with kidney disease [29,30,31]. However, there are few studies looking at fibre intake and the burden of other uraemic symptoms, especially when using a validated tool for kidney disease. Furthermore, our study is the first to examine the relationship between fibre intake, QOL and physical strength in patients with kidney disease.

There was a relationship between serum albumin and patient-focused outcomes, where lower serum albumin was associated with higher symptom burden, lower quality of life and poorer physical ability. Albumin is a strong predictor of mortality in patients with kidney disease [32,33]. However, its association with symptom burden and physical ability has only been studied in non-kidney disease populations or retrospective studies in kidney disease populations [34,35,36]. The exact mechanism by which this transpires in unclear, but it could be attributed to the systemic inflammatory state in kidney disease, where albumin is a negative acute phase reactant.

The transplant patients consistently performed better in the assessment of physical ability and PROMs. This likely reflects their lower uraemic status, with significantly lower PBT concentrations compared with dialysis patients. Interestingly, we found that the transplant patients had a higher median daily fibre intake of 19.3 g/day, compared with PD (11.8 g/day) and HD groups (13.0 g/day), although no group and few individuals achieved the daily recommended amount of 25–30 g/day. Nonetheless, the superior performance of transplant patients compared with dialysis patients could be attributed to both a lower uraemic toxin burden as well as better fibre intake.

There are multiple limitations to our study. Given its cross-sectional nature, we can only determine association. Therefore, the interplay between PBTs, patient-focused outcome measures and fibre intake needs to be further evaluated in interventional studies with a larger sample size, particularly a randomised controlled trial. Assessment of diet history can be challenging. In both clinical practice and research, it can be subject to inherent biases associated with self-reporting and recall bias. However, pictorial aids and other prompts during historical entry (e.g., portion size) into the smart phone application did allow us to gather a diet history that was consistent on initial screening at least with dietitians’ assessments. Lastly, our cohort of patients had an unusually high QOL (median EQ-5D-5L index value of 0.96, 0.93 and 1.0 for PD, HD and transplant patients, respectively) compared to other studies [37]. This may explain the lack of association found between PBT, fibre intake and QOL. Further studies in larger cohorts or using another validated QOL PROM instrument may help evaluate this relationship better.

Despite these limitations, our study is the first to demonstrate an association of PBTs with symptom burden and QOL using PROM tools known to be validated for kidney disease (IPOS-renal and EQ-5D-5L). The findings indicate that PBTs could play a role in uraemic symptomatology and may provide the missing link between poorer patient outcomes and uraemia. Our study also revealed an interesting interplay between PBTs, patient-focused outcomes and fibre.

## 4. Conclusions

Prevailing concentrations of PBTs were associated with poorer patient-focused outcomes, including symptom burden and physical strength, in patients with kidney disease. In turn, fibre intake was closely associated with these findings, with lower intake reflecting higher PBT concentrations and poorer outcomes. These preliminary results warrant additional, longitudinal studies to determine if fibre might be a potential therapeutic intervention to reduce serum PBTs and improve outcomes in CKD patients.

## 5. Methods

We conducted a cross-sectional study of three groups of patients on kidney replacement therapy: prevalent HD, PD and kidney transplant patients. We chose kidney transplant patients as a comparable comparator to the dialysis patients as they are presumed to have a lower uraemic burden with low PBT levels. This study was carried out at two large tertiary nephrology centres in metropolitan Victoria, Australia. The study was approved by the local human research ethics committee (reference number HREC/94965/EH-2023-376581 v3). All patients were recruited when attending routine outpatient and dialysis appointments. At the time of study visit, demographic details, medical history, medication list and social history were collected.

The inclusion criteria included age > 18 years, prevalent PD and HD patients (that had been on that modality for >3 months) and prevalent kidney transplant patients (transplant > 3 months). The exclusion criteria included those who were pregnant, had cognitive impairment and/or learning differences or those who were not expected to be on their dialysis modality for more than 12 months.

### 5.1. Diet History

We collected a three-day diet history (two weekdays and one weekend day) and calculated the daily average fibre intake using the Easy Diet Diary smart phone application Xyris Software Australia Pty Ltd., High Gate Hill, Australia, Version 7.2.2 Build 380). This application has been used in routine clinical practice and epidemiological research with good validity, feasibility and usability in kidney and non-kidney disease populations [38,39,40,41]. We trialled this application on three healthy volunteer diet diaries and compared results with a kidney unit dietician. Results demonstrated a coefficient of variation of <0.08.

### 5.2. Patient-Reported Outcome Measures (PROMs)

We assessed symptom burden using the IPOS-renal questionnaire that assesses the severity of uraemic symptoms, psychological stressors, communication issues and satisfaction with medical treatment. It has been validated in dialysis patients with good re-test reliability, internal consistency and translation into other languages across the world [42,43]. We assessed QOL using the EQ-5D-5L survey, which measures health-related QOL across five dimensions: mobility, self-care, usual activities, pain and anxiety/depression. The index value is where the five dimensions of QOL are summarised by a single number normalised for the Australian population using a standardised valuation exercise, where 1 = the value of full health and 0 is death [44]. The visual analogue scale is a quantitative measure of patients’ perception of their overall health, where 100% is the best health imaginable. We chose this tool because it has been validated in kidney disease with similar validity to other measures such as kidney disease quality of life (KDQOL) [45]. It is simple to use and less time-consuming than other measures, thus avoiding survey fatigue.

### 5.3. Physical Ability

Handgrip strength (HGS) was measured using a dynamometer. This is an indicator of physical functioning and a predictor of morbidity and mortality [46,47]. It is also associated with the frailty phenotype in older adults and is a surrogate marker for functional status and protein–energy metabolism in patients with kidney disease [48,49].

Lower body strength was assessed using the timed up and go (TUG) test, where patients are asked to stand up from a chair, walk 3 m, turn around, walk back to the chair and sit down at their normal pace. It has been used in a wide range of populations, including kidney disease, to assess functional mobility. It has validity and re-test reliability in non-kidney disease older populations but is also associated with frailty and fractures in HD patients [50,51,52].

### 5.4. Pathology

Serum samples (15–20mL of blood in 2 serum tubes and 1 EDTA tube) for PBTs were collected from haemodialysis patients before dialysis using their arterio-venous fistula or permanent dialysis catheter at their mid-week dialysis session. Serum samples (as above) for PBTs were collected from PD and kidney transplant patients during their routine adequacy assessments and outpatient clinical appointments, respectively. IS and PCS were measured using methods defined previously by Calaf and team using a Shimadzu High-Performance Liquid Chromatography system where the PBTs were quantified using fluorescence detection [53].

### 5.5. Statistics

The data were tabulated in Microsoft Excel (Microscoft 365, USA 2025, Version 16.101.3) and analysed using GraphPad Prism software (version 10.5.0, Massachusetts, USA). A power analysis was conducted based on the study by Bammens where a statistically significant correlation between PCS levels and uraemic symptoms was found in 30 PD patients [11]. Taking this into account, we estimated a sample size of n = 30 to detect an association between PBTs and patient-focused outcome with 90% power and a 2-sided alpha of 0.05. Normality of the data was determined by the Shapiro–Wilk and Kolmogorov–Smirnov tests.

Correlation coefficients were used to examine the strength of the relationship between two variables: Pearson’s correlation coefficient for normally distributed data, and Spearman’s rank correlation coefficients for non-parametric data. If there was a significant correlation, multiple linear regression analysis was performed to account for confounding variables that were clinically relevant and could impact function and symptoms (e.g., age, gender and co-morbidities). We used Fisher’s exact test to compare categorical baseline patient characteristics (e.g., gender, diabetes, comorbidity) between the three patient groups. Comparison of continuous variables between the three groups was compared using one-way ANOVA for normally distributed data and the Kruskal–Wallis test for non-parametric data. Tukey’s multiple comparisons post hoc test was applied when comparisons were made between groups. A *p*-value < 0.05 was considered statistically significant.

## Figures and Tables

**Figure 1 toxins-17-00548-f001:**
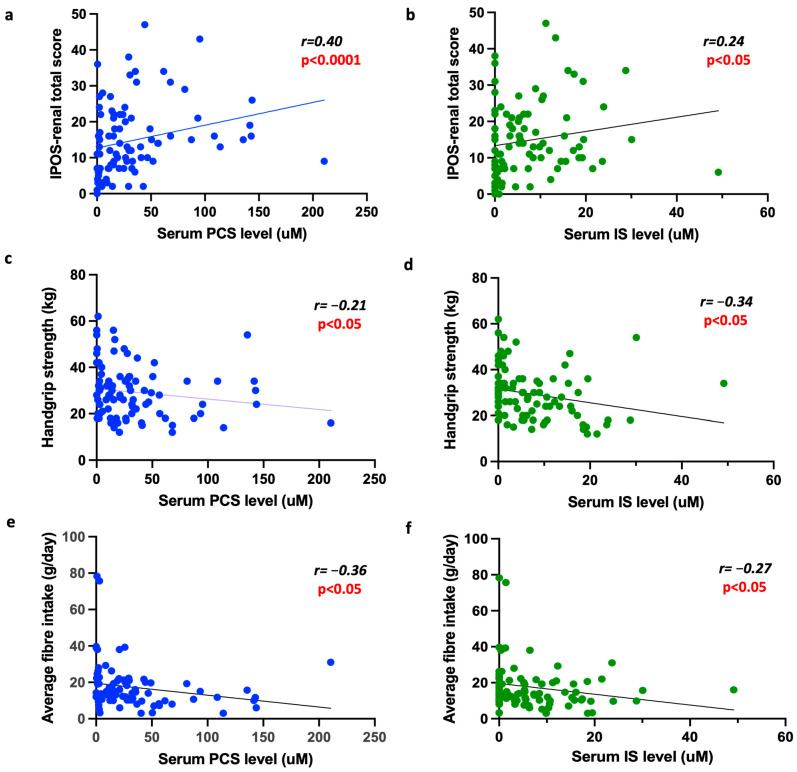
The association between PBT levels, patient-focused outcomes and fibre intake. PBT levels were positively associated with IPOS-renal scores (**a**) for PCS and (**b**) for IS. PBT levels were negatively associated with handgrip strength (**c**) for PCS and (**d**) for IS. Finally, PBT levels were negatively associated with fibre intake (**e**) for PCS and (**f**) for IS.

**Figure 2 toxins-17-00548-f002:**
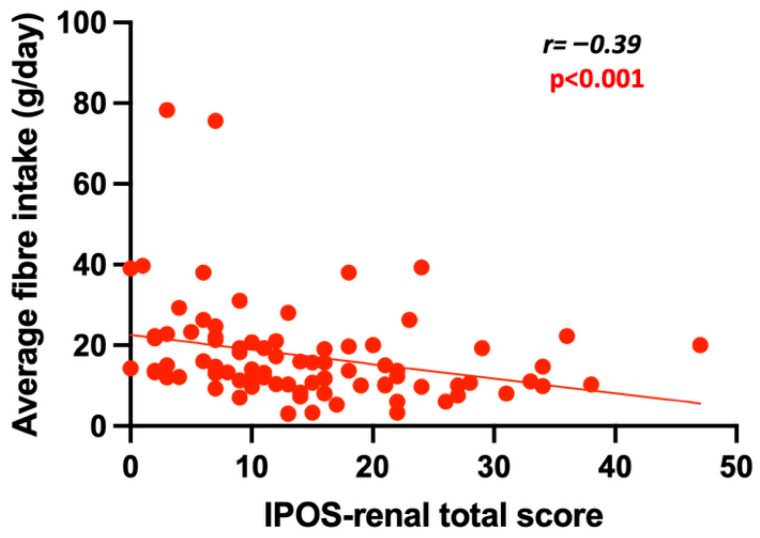
The association between fibre intake and the total IPOS-renal score. Lower fibre intake was associated with a higher symptom burden measured by the IPOS-renal.

**Table 1 toxins-17-00548-t001:** Baseline characteristics between the three patient groups.

	All (n = 90)	PD (n = 30)	HD (n = 30)	Transplant (n = 30)	*p* Value
Age (years) [Median (IQR)]	63 (54–74)	59 (45–74)	72 (59–80)	59 (54–67)	<0.05
Male sex [N (%)]	59 (66%)	17 (57%)	20 (67%)	22 (73%)	NS
Aetiology of kidney disease [N (%)]					
• Diabetes and hypertension	25 (28%)	5 (17%)	14(47%)	6 (20%)	<0.05
• Glomerulonephritis/vasculitis	27 (30%)	11 (36%)	4 (13%)	12 (40%)	
• Other	38 (42%)	14 (47%)	12 (40%)	12 (40%)	
Prevalence of diabetes	31 (34%)	6 (20%)	14 (47%)	11 (37%)	NS
Charleson co-morbidity score (CC score) [Median (IQR)]	5 (3–7)	4 (3–6) ^a^	7 (5–8)	4 (3–6) ^b^	<0.01
BMI (kg/m^2^) [Median (IQR)]	27 (23–29)	27 (24–29)	25 (22–29)	27 (23–31)	NS
Mobility inside [N (%)]					
• Independent	82 (91%)	29 (97%)	24 (80%)	29 (97%)	NS
• Stick	4 (5%)	1 (3%)	3 (10%)	0 (0%)	
• Frame/walker	2 (2%)	0 (0%)	2 (7%)	0 (0%)	
• Wheelchair	2 (2%)	0 (0%)	1 (3%)	1 (3%)	
Mobility outside [N (%)]					
• Independent	76 (84%)	27 (90%)	20 (67%) ^c^	29 (97%)	<0.001
• Stick	4 (5%)	1 (3%)	3 (10%) ^c^	0 (0%)	
• Frame/walker	8 (9%)	2 (7%)	6 (20%) ^c^	0 (0%)	
• Wheelchair	2 (2%)	0 (0%)	1 (3%)	1 (3%)	
PROM tool					
EQ-5D-5L index value Median (IQR)	0.96 (0.89–1)	0.96 (0.89–0.98)	0.93 (0.79–0.98)	1 (0.95–1)	NS
EQ-5D-5L visual analogue scale Median (IQR)	80 (59–90)	70 (63–88)	65 (50–85) ^d^	85 (80–90)	<0.01
IPOS-renal total score Median (IQR)	13 (7–21)	16(11–23) ^e^	13 (9–21)	7 (3–16)	<0.05
Physical ability					
HGS Kg) Median (IQR)	28 (20–34)	24 (18–31) ^f^	26 (18–34) ^g^	33 (28–45)	<0.001
TUG (s) Median (IQR)	10.4 (8.0–12.4)	11.3 (9.1–14.1)	11.6 (8.3–17.9) ^h^	9.4 (7.5–10.6)	<0.01
PBTs					
PCS (µM) Median (IQR)	20.7 (3.58–40.8)	53.4 (12.8–98.6)	25.1 (15.5–33.4)	4.4 (1.2–15.1)	<0.0001
IS (µM) Median (IQR)	5.2 (0.42–11.1)	9.0 (5.1–15.4) ^i^	8.4 (5.0–16.4) ^i^	0 (0–0.59)	<0.01
Fibre intake					
g/day Median (IQR)	13.8 (10.3–20.8)	11.8 (8.3–15.7)	13 (9.9–19.7)	19.3 (13.5–26.3) ^j^	<0.05

Key: CC score: ^a^
*p* < 0.001 versus PD, ^b^
*p* < 0.01 versus transplant; Mobility outside: ^c^
*p* < 0.01 versus transplant; EQ-5D-5L visual analogue scale: ^d^
*p* < 0.01 versus transplant; IPOS-renal total score: ^e^
*p* < 0.01 versus transplant; HGS: ^f^
*p* < 0.001 versus transplant, ^g^
*p* < 0.01 versus transplant; TUG: ^h^
*p* < 0.051 versus transplant; IS: ^i^
*p* < 0.0001 versus transplant; Fibre intake: ^j^
*p* < 0.05 vs. HD and PD. Abbreviations: HD: Haemodialysis; PD: Peritoneal dialysis; BMI: Body Mass Index; PROM: Patient-reported outcome measures; IPOS-renal: Integrated palliative outcome score (IPOS-renal); HGS: Handgrip strength; TUG: Timed up and go; PBTs: Protein-bound uraemic toxins; PCS: *p*-cresyl sulfate; IS: Indoxyl sulfate.

**Table 2 toxins-17-00548-t002:** Associations between patient-focused outcome measures and PCS in a univariable and multivariable analysis.

PCS				
	Univariable Analysis	Multivariable Analysis (Model 1)	Multivariable Analysis (Model 2)	Multivariable Analysis (Model 3)
PROM tool				
EQ-5D-5L Index value (r, 95% CI)	r = −0.18 (−0.38–0.03) *p* = NS	N/A	N/A	N/A
EQ-5D-5L VAS (r, 95% CI)	r = −0.16 (−0.36–0.06) *p* = NS	N/A	N/A	N/A
IPOS-renal total score (r, 95% CI)	r = 0.40 (0.20–0.56) *p* < 0.0001	*p* < 0.05	*p* < 0.05	No further significance
Physical ability				
HGS (Kg) (r, 95% CI)	r = −0.21 (−0.40–0.005) *p* < 0.05	No further significance	No further significance	No further significance
TUG (seconds) (r, 95% CI)	r = 0.34 (0.13–0.51) *p* < 0.05	No further significance	No further significance	No further significance

Model 1: Adjusted for age, gender, BMI and the Charlson Co-morbidity Index. Model 2: Adjusted for RRF (urine output). Model 3: Adjusted for albumin, protein and fat intake. Abbreviations: PROM: Patient-reported outcome measures; IPOS-renal: Integrated palliative outcome score (IPOS-renal); HGS: Handgrip strength; TUG: Timed up and go; PCS: P-cresyl sulfate, NS: Not significant, N/A: Not applicable.

**Table 3 toxins-17-00548-t003:** Associations between patient-focused outcome measures and IS in a univariable and multivariable analysis.

IS				
	Univariable Analysis	Multivariable Analysis (Model 1)	Multivariable Analysis (Model 2)	Multivariable Analysis (Model 3)
PROM tool				
EQ-5D-5L Index value (r, 95% CI)	r = −0.17 (−0.37–0.05) *p* = NS	N/A	N/A	N/A
EQ-5D-5L VAS (r, 95% CI)	r = −0.2 (−0.40–0.17) *p* = NS	N/A	N/A	N/A
IPOS-renal total score (r, 95% CI)	r = 0.24 (0.03–0.43) *p* < 0.05	No further significance	No further significance	No further significance
Physical ability				
HGS (Kg) (r, 95% CI)	r = −0.34 (−0.52–0.14) *p* < 0.05	*p* < 0.05	*p* < 0.05	No further significance
TUG (seconds) (r, 95% CI)	r = 0.30 (0.1–0.49) *p* < 0.01	No further significance	No further significance	No further significance

Model 1: Adjusted for age, gender, BMI and the Charlson Co-morbidity Index. Model 2: Adjusted for RRF (urine output). Model 3: Adjusted for albumin, protein and fat intake. Abbreviations: PROM: Patient-reported outcome measures; IPOS-renal: Integrated palliative outcome score (IPOS-renal); HGS: Handgrip strength; TUG: Timed up and go; IS: Indoxyl sulfate, NS: Not significant, N/A: Not applicable.

**Table 4 toxins-17-00548-t004:** Associations between patient-focused outcome measures and fibre intake in a univariable and multivariable analysis.

	Fibre Intake	
	Univariable Analysis	Multivariable Analysis *
PROM Tool		
EQ-5D-5L Index value (r, 95% CI)	r = 0.19 (−0.3–0.40) *p* = NS	N/A
EQ-5D-5L VAS (r, 95% CI)	r = 0.21 (−0.01–0.42) *p* = NS	N/A
IPOS total score (r, 95% CI)	r = −0.39 (−0.56–0.19) *p* < 0.001	*p* < 0.05
Physical ability		
HGS (r, 95% CI)	r = 0.11 (−0.11–32) *p* = NS	N/A
TUG (r, 95% CI)	r = −0.28 (−0.47–0.06) *p* < 0.05	No further significance

* Adjusted for age, gender, socio-economic status, BMI and Charlson Co-morbidity Index. Abbreviations: PROM: Patient-reported outcome measures; IPOS-renal: Integrated palliative outcome score (IPOS-renal); HGS: Handgrip strength; TUG: Timed up and go, NS: Not significant, N/A: Not applicable.

**Table 5 toxins-17-00548-t005:** Associations between patient-focused outcome measures and serum albumin in a univariable and multivariable analysis.

Serum Albumin				
	Univariable Analysis	Multivariable Analysis (Model 1)	Multivariable Analysis (Model 2)	Multivariable Analysis (Model 3)
PROM tool				
EQ-5D-5L Index value (r, 95% CI)	r = 0.23 (0.01–0.43) *p* < 0.05	No further significance	No further significance	N/A
EQ-5D-5L VAS (r, 95% CI)	r = 0.17 (−0.05–0.38) *p* = NS	No further significance	No further significance	N/A
IPOS-renal total score (r, 95% CI)	r = −0.32 (−0.51–0.11) *p* < 0.01	*p* < 0.05	No further significance	*p* < 0.01
Physical ability				
HGS (Kg) (r, 95% CI)	r = 0.32 (0.11–0.51) *p* < 0.01	*p* < 0.01	*p* < 0.05	*p* < 0.01
TUG (seconds) (r, 95% CI)	r = −0.39 (−0.56–0.19) *p* < 0.001	No further significance	No further significance	No further significance

Model 1: Adjusted for age, gender, BMI and the Charlson Co-morbidity Index. Model 2: Adjusted for RRF (urine output). Model 3: Adjusted for protein and fat intake. Abbreviations: PROM: Patient-reported outcome measures; IPOS-renal: Integrated palliative outcome score (IPOS-renal); HGS: Handgrip strength; TUG: Timed up and go, Not significant, N/A: Not applicable.

## Data Availability

The original contributions presented in this study are included in the article. Further inquiries can be directed to the corresponding authors.

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
