# Peer review of "Well-Being, Protein-Bound Toxins, and Dietary Fibre in Patients with Kidney Disease: Have We Been Missing the Obvious?"

_toxins, 2025, doi:10.3390/toxins17110548_

Round 1
Reviewer 1 Report
Comments and Suggestions for Authors
The manuscript entitled: Well-Being, Protein-Bound Toxins, and Dietary Fibre in PD Patients: Have We Been Missing the Obvious?, sets out to explore the associations between protein-bound uraemic toxins, fibre in-take and patient-focused outcomes in patients on kidney replacement therapy. In general the manuscript is very (too) descriptive and text could be reduced significantly.
specific comment:
-The title refers to PD patient, while also HD and transplanted patients are included, maybe better to refer to KRT
-Remove from the introduction “Fibre intake has been a recent innovation to reduce PBT production in kidney failure.”, not clear why this is innovative.
-Why should the aim be stated separately at the end of the introduction? Better to incorporate.
-Replace “renal” by “kidney” whenever possible
-Line 89: please write coefficient one word
-Methods: HPLC is a method to separate metabolites, detection occurs by the UV or fluorescence detector, please add.
-Which normality test was used?
-Footnote to the table 1, next to instead of below each other
-Please explain the used abbreviations in the footnote of the tables
-It is redundant to show similar results twice (see table 2 and figure 1).
Legend to figure 1: please refer to positive or negative correlations instead of “higher level … are associated with higher levels…” and just state what is illustrated in the figures, interpretations should be stated in the text. In the legend: Association of UTs levels with (a and b) IPOS-renal score; (c and d)handgrip strength; …
It is unnecessary to described the same findings twice, second part of the sentence is redundant, “There was a significant negative correlation between HGS and PCS, where higher PCS levels were associated with a lower HGS in a univariable analysis (r=-0.21, p<0.05)”. same further on.
-Discussion
-Line 258: add ref of Bammens et al.
-In general, the discussion should include a better framing of the selected parameters. Could authors give reasons for the unusual QoL score?
Reviewer 2 Report
Comments and Suggestions for Authors
This manuscript investigates the association between protein-bound uremic toxins (PBTs), dietary fibre intake, and patient-reported outcomes in patients on kidney replacement therapy (hemodialysis, peritoneal dialysis, and kidney transplant). Using validated PROMs (IPOS-Renal and EQ-5D-5L), as well as physical function measures (handgrip strength and TUG test), the authors identify significant associations between higher PBT levels, higher symptom burden, and lower physical function. Additionally, lower fibre intake correlates with higher toxin levels and poorer outcomes. The study addresses a highly relevant and timely topic in nephrology — the interplay between uraemic toxins, gut-derived metabolites, and patient well-being. The cross-sectional approach and use of validated PROMs are commendable. However, the current manuscript has several major methodological, analytical, and interpretative limitations that must be addressed before it can be considered for publication.
- The cross-sectional design precludes causal inference, yet several conclusions are phrased in a way implying directionality (“fibre intake influenced this relationship,” “fibre may be a therapeutic intervention”). The authors should carefully rephrase all such statements and acknowledge that the observed correlations cannot establish causality.
- The sample size (n=90) is modest for multivariable regression, especially given multiple confounders (age, comorbidity, BMI, SES). Power calculations are not provided and should be included to justify the adequacy of statistical analysis.
- Regression models are insufficiently detailed. The manuscript states that multiple linear regressions were performed but does not specify model diagnostics (e.g., residual distribution, collinearity, or inclusion criteria for variables).
- Correlation coefficients are interpreted as “strong,” though values around 0.3–0.4 are only moderate. Overinterpretation should be avoided.
- Adjustment for multiple testing is not discussed despite numerous comparisons, increasing the risk of Type I error.
- The use of median and IQR suggests non-parametric distributions, yet ANOVA and Pearson’s r are reported in some instances. Methods should match data distribution.
- The three-day dietary record via a smartphone app is convenient but prone to recall bias and underreporting. The authors should explicitly discuss the potential measurement bias and provide evidence for validation in renal populations.
- Fibre intake estimation should ideally be normalized for energy intake (e.g., g/1000 kcal) to account for differences in dietary patterns.
- While correlations between PBTs and IPOS-Renal scores are interesting, no clear biological mechanism linking specific toxins to symptom burden is proposed. The discussion should explore plausible pathways (neuromodulation, systemic inflammation, gut-derived metabolites).
- The lack of association with EQ-5D-5L is underexplored. This discrepancy deserves interpretation (e.g., EQ-5D may lack sensitivity to uraemia-specific symptoms).
- The claim that “PBTs could be a quantifiable marker of uraemic toxicity” is overstated given the study’s observational design.
- Important confounders such as residual kidney function, inflammatory status (CRP), nutritional status, or albumin levels are not incorporated in regression analyses. These could significantly affect both toxin levels and well-being.
- Medication use (e.g., phosphate binders, antibiotics, probiotics) can also influence PBT production and should be discussed as limitations.
- Specify units consistently (e.g., µM vs μmol/L).
Reviewer 3 Report
Comments and Suggestions for Authors
The comments follow throughout the attached document.

Round 2
Reviewer 1 Report
Comments and Suggestions for Authors
Authors addressed my comment in a satisfactory manner.
Reviewer 2 Report
Comments and Suggestions for Authors
The article has been revised and, in its current form, can be considered by the editor for publication.
Reviewer 3 Report
Comments and Suggestions for Authors
Significant improvements have been made to the manuscript, and can be published.
In line 39, I think it would be better if uric acid were added.